# Real-Time Metasurface Sensor for Monitoring Micropoisons in Aqueous Solutions Based on Gold Nanoparticles and Terahertz Spectroscopy

**DOI:** 10.3390/s22031279

**Published:** 2022-02-08

**Authors:** Amir Abramovich, Yossi Azoulay, David Rotshild

**Affiliations:** Department of Electrical and Electronic Engineering, Ariel University, Ariel 40700, Israel; yossiazoulay8@gmail.com (Y.A.); davidrot@ariel.ac.il (D.R.)

**Keywords:** metasurface, double split rectangular resonator, gold nanoparticles, micropoisons in aqueous solutions, frequency-domain spectroscopy

## Abstract

Proof of concept of a new real-time metasurface sensor for micropoison monitoring in aqueous solutions is proposed in this study. The sensor comprises a perfect absorber metasurface and gold nanoparticle layer on the front side of it. Frequency-domain terahertz spectroscopy system was used to measure the resonance frequency shift due to the presence of the micropoison. The perfect absorber metasurface sensor was fabricated using a double-sided FR4 substrate printed board circuit, which is very inexpensive. A significant increase in the metasurface sensor sensitivity was achieved by adding a gold nanoparticle layer to the gap of the double split rectangular resonator on the front side of the metasurface sensor.

## 1. Introduction

Contamination of drinking water by micropoisons is a well-known issue. There are many water micropoison contamination sources, including naturally occurring sources and local land-use practices such as fertilizers, pesticides, organophosphates, and concentrated feeding operations. Manufacturing processes and sewer overflows or wastewater releases are contamination sources [1]. The wide use of organophosphates in agriculture and industry in the last few decades caused contamination in groundwater and water infrastructures.

Some of these organophosphates are harmful to humans and animals, even in negligible quantities since they can cause cumulative damage. In addition, organophosphates interfere with healthy neurodevelopment, causing behavioral and cognitive problems [2]. Thus, real-time detection and recognition of micropoisons in drinking water are required to protect the world’s population.

There are chemical detection methods such as HPLC [3], ELISA [4], and GCMS [5] on the one hand, and there are optical methods such as mid-infrared (MIR) spectroscopy, attenuated total reflection (ATR) crystal, and fiber-optic ATR on the other hand [6,7]. However, the chemical measurement systems are costly, require complex material preparations, and take too long to receive the results. On the other hand, the spectroscopic methods [8,9] are fast and can be done in real time. However, the FTIR-ATR method is less sensitive and requires special coatings for the ATR crystal [10,11].

This study proposes a new detection concept for real-time drinking water quality monitoring. The measurement system is based on a combination of an advanced CW frequency-domain terahertz spectrometer and a perfect absorber metasurface (MS) sensor coated with a thin layer of gold nanoparticles. The perfect absorber metasurface [12,13] was fabricated on double-sided FR4 substrate using printed board circuit (PCP) technology, which is very inexpensive (see Figure 1) [14]. The resonance frequency, *f*_res_, of the perfect absorber is determined according to the geometrical metal shape dimensions and the dielectric constant of the substrate (see Equations (1)–(4)). The designed resonance frequency of the perfect absorber MS shifts slightly from its designed value due to the presence of micropoisons in proximity to the perfect absorber MS, especially near the gap of the double split rectangular resonator (DSRR) (see Figure 1a). The higher the micropoison concentration in the water, the more significant the shift in the resonance frequency, *f*_res_, of the MS sensor. The resonance frequency, *f*_res_, shift due to the presence of micropoisons is measured using a high-resolution compact terahertz CW frequency-domain spectrometer. 

Adding a gold nanoparticle coating on the MS surface, especially in the gap of the DSRR, causes the micropoisons such as OPs to adhere to the MS, increasing the detection sensitivity significantly [11]. In this study, we present a full proof of concept of malathion detection using a perfect absorber MS manufactured by well-known and very inexpensive printed circuit board (PCB) technology. The spectral measurements were carried out using a high-resolution terahertz spectroscopy system [15].

## 2. Metasurface Design, Simulations, and Experimental Setup

There are many geometries to realize perfect absorber MS, including fishnet, split rectangular, and cut wire [16,17]. In this work, we used the double split rectangular resonator (DSRR) geometry on the front side of the FR4 substrate and cut wire geometry on the back side.

The proposed MS sensor detection mechanism is based on shifting of the MS sensor resonance frequency, *f*_res_, due to the presence of micropoisons near the gap of the DSRR of the MS, rather than the MIR spectroscopic signature of the micropoison [18]. Furthermore, significant improvement in the MS sensor sensitivity can be achieved by adding a thin layer of gold nanoparticles on the front side of the MS sensor, especially in the gap of the DSRR [11].

### 2.1. Metamaterial Design and Simulations

The sensor design is based on a perfect absorber metasurface structure. The unit cell of the perfect absorber consists of a double split rectangular resonator (DSRR) [19] at the front side and a cut wire (CW) [20] at the back side of the metasurface structure. Figure 1 shows the perfect absorber geometry and its equivalent circuit.

The double split rectangular resonator (DSRR) can be modeled as an array of parallel *LC* resonance creating a perfect absorber metasurface. The equivalent *LC* model of the MS sensor is shown in Figure 1d. The resonance frequency, *f*_res_, is determined according to the well-known resonance formula:(1)$$\omega =\frac{1}{\sqrt{LC}}\Rightarrow f_{res}=\frac{1}{2\pi \sqrt{LC}}$$

The inductance, *L*, is due to the two current loops of the DSRR rectangles and is given approximately by [21]:(2)$$L\approx \mu_{0}r_{m}\left[ln\left(\frac{8r_{m}}{c+d}-0.5\right)\right]$$
where *c* and *d* are the DSRR strip width and copper thickness, respectively. *r_m_* = (w *+ l*)/2 is the approximate value of the loop radius. The permeability constant *µ*_0_ = 1.2566 µh/m; *µ_r_* = 1. The capacitance, *C*, is composed of the split capacitance, *C_split_*, and the surface capacitance, *C_surf_*, in series (see Figure 1d) [21]. Approximate formulas for the values of the split capacitances are given in [19]:(3)$$C_{split}=\varepsilon_{0}\left(\frac{cd}{s}+\frac{2\pi d}{ln\left(\frac{2.4d}{c}\right)}\right)$$
(4)$$c_{surf}=\varepsilon_{0}\frac{2d}{\pi }ln\left(\frac{2r_{m}}{s}\right)$$
where is the DSRR gap. The vacuum permittivity is ε_0_ = 8.8541878 pF/m, and the permittivity of FR4 is ε*_r_* = 4 at 100 GHz. The DSRR unit cell design parameters are given in Table 1.

Based on the unit cell parameters given in Table 1 and Equations (1)–(4), the resonance frequency, *f*_res_, was calculated and found to be on the order of 125–145 GHz, depending on the estimated value of *r_m_*.

The physical dimensions of the fabricated MS sensor are 135 mm × 135 mm, and the total number of unit cells was 21,960. A microscopic photo of the MS is shown in Figure 1e.

### 2.2. Experimental Setup

The CW frequency-domain spectroscopic terahertz system is based on the TeraScan CW terahertz spectrometer of Toptica Inc. [17,21].

The terahertz spectrometer has two programmable distributed feedback diode (DFB) lasers that generate two adjacent frequencies, λ_1_ and λ_2_, around 1.5 μm. A 50% fiber-optic coupler/splitter is used to couple the two laser beams λ_1_ and λ_2_, with λ_1_ + λ_2_ exiting at two split fibers. The difference frequency generation (DFG) process is realized by a state-of-the-art InGaAs photomixer (TOPTICA Photonics AG, Munich, Germany) generating the terahertz radiation. The second InGaAs photomixer is used for the detection of the terahertz radiation (see Figure 2). Applying a bias voltage to the metal electrodes of the Tx photomixer generates a photocurrent that oscillates at the beat frequency (see Figure 2). A bow-tie antenna on the photomixer emits an electromagnetic wave at the terahertz difference frequency (see Tx in Figure 2) [15]. Rx, a second photomixer, is used at the receiver side to detect the transmitted terahertz signal using look-in detection [13,21]. The advantages of this technique include high spectral resolution selectivity, a superior dynamic range of up to 100 dB, and a substantial spectral range of 50–1200 GHz. Figure 2 shows the block diagram of the whole spectroscopic system [21]. This spectrometer has a high spectral resolution of better than 10 MHz and high SNR due to the coherent detection. The terahertz radiation departing from the transmitter is linearly polarized. In this experimental work, the terahertz radiation power at the 100–200 GHz range is approximately 3.5 microwatts, and the SNR in this range is −80 dB [21]. This spectrometer has a high spectral resolution of better than 10 MHz and high SNR due to the coherent detection. The terahertz radiation departing from the transmitter is linearly polarized. In this experimental work, the terahertz radiation power at the 100–200 GHz range is approximately 3.5 microwatts, and the SNR in this range is −80 dB [21].

The MS sensor was placed at the focal point of a configuration of four off-axis parabolic mirrors, where the incident beam is normal to the MS, as shown in Figure 2. All measurements were performed with a half-inch aperture. The relative measurement method was used in this experimental setup. At first, measurements were performed only for a clear aperture as a reference; then, they were performed for the MS sensor without the OP (pure MS). The polarity of the terahertz spectroscopy system is perpendicular to the scheme plane. Therefore, the MS was positioned along the *Y*-axis (see Figure 1a). In addition to signal processing, the relation between measurements provided the transmission characteristic of the MS sensor. After achieving the original transmission characteristic, malathion pesticide was dropped on the MS sensor. Then, the measurement procedure was repeated, for two different periods of time, to investigate the MS sensor response to the presence of malathion.

## 3. Experimental Results

Simulation results of the designed MS were obtained using CST 3D electromagnetic simulation code [22]. Figure 3 shows the transmittance of the designed perfect absorber MS shown in Figure 1 for the parameters given in Table 1. The spike effects in the measurement curve (solid black line in Figure 3) are due to measurement procedure and calibration to the reference measure, which is typical for measurement using the TeraScan CW terahertz spectrometer of Toptica [21].

The influence of malathion on the terahertz spectral results is given in Figure 4. The black curve describes the transmittance, T, of the perfect absorber metamaterial frequency in an aqueous environment without toxins, the red curve represents the MS absorber frequency response in an aqueous environment with malathion, and the green curve shows the frequency of absorption after one hour in free space outside the toxin environment. Malathion is known as an evaporating material. These experimental results provide an encouraging and real-time method for malathion detection in an aqueous environment. The detection is based on the resonance frequency shift of the perfect absorber MS due to the presence of malathion.

## 4. Discussion

The simulation and the experimental results proved that the MS sensor described above (see Figure 1) could be used as a real-time sensor for micropoison detection in aqueous solutions. Further improvement in the detection sensitivity can be achieved by adding a thin layer of gold nanoparticles in the gap of the DSRR, as shown in Figure 5.

In recent years, nanoparticles have played an essential role in manufacturing sophisticated products [23,24,25]. Products such as inductors, capacitors, and tuned circuits have low-resistance conductors based on nanotechnology. In the biological and imaging industry, nanotechnology has significantly improved the quality of the products [24,25]. Nowadays, this technology is available, and the shelves, components, and materials are commercially available. For example, we can increase the sensitivity of detection technologies such as “ATR FTIR” measurement [13]. Furthermore, the nanoparticle technology enables a new concept of preconcentration using ATR crystal coatings of gold nanoparticles. In this study, we propose the addition of a thin layer of gold nanoparticles in the gap of the DSRR, attracting the malathion molecules, as shown in Figure 5.

Figure 6 shows the effect of gold ATR coatings on the detection of malathion [11]. There are two important conclusions: The first is that metal nanoparticles attract malathion and probably other micropoisons in aqueous solutions. The second is that the attraction of the micropoison to the gold nanoparticles increases the sensitivity and recognition by ATR FTIR measurement [11]. Based on those conclusions, we simulated the influence of adding gold nanoparticles to the DSRR gap on the sensor sensitivity.

There are two resonances in the designed MS: one in X polarization and the second in Y polarization, as shown in Figure 7. The resonance frequency is 132 GHz for X polarization and 136 GHz for Y polarization, as shown by the solid blue and black lines, respectively. This includes the addition of a gold nanoparticle (GNP) coating. These simulation results are in excellent agreement with the approximate calculation of Equations (1)–(4). Furthermore, the adhesion of the OP to the gold NP caused a shift in the resonance frequency, as can be seen by the dashed red line only for Y polarization, thus significantly increasing the sensitivity of the MS to the presence of OP in the aqueous solution. No influence of the particles in X polarization was observed. The higher the concentration of the OP in the solution, the more significant the resonance frequency shift will be (in Y polarization). Figure 5a shows simulation results of the electric field intensity distribution on the MS. Note that the maximum intensity is at the split edges. Thus, adding the GNP near the split edges is expected to maximize the sensor’s sensitivity, as can be seen in Figure 5b. Figure 6c shows the adhesion of malathion particles to the NGP, changing the resonance frequency of the MS, as shown in the dashed red line of Figure 8. An estimation, based on CST simulation, of the resonance frequency shift as a function of the OP concentration in the solution is shown in Figure 8.

Figure 5a represents the central concept—the gold nanoparticles on the DSRR gap. In Figure 5b, we can see the gold nanoparticles in the gap between the copper edges. Figure 5c shows the simulation of the toxic malathion molecules attracted by the gold nanoparticles. 

Figure 7 represents the CST simulation results of MS with gold (red dashed and green dashed lines) and without the covered gold DSRR (black line and blue line). The simulations are in X and Y polarization, and the results clearly show the same transmittance and absorption frequency.

Figure 8 represents the simulation results of the toxic aqueous environment and gold nanoparticles. We can see clearly that the X polarization (the red dashed and green lines) is without change in the absorption frequency, while in the Y polarization (the black line and red lines), we can see the difference and the shift in absorption frequency.

## 5. Conclusions

The MS coated with gold nanoparticles, whose results are presented in Figure 8, proves three critical principles. The first is that metal nanoparticles attract micropoisons in aqueous solutions. We can see these results in Figure 5. The second is that the micropoison attracted to the gold nanoparticles can be detected and recognized by the MS in real-time measurement, and the last is that it is cheap and easy to use and manufacture. Thus, this new concept can control and monitor toxic water or aqueous solutions. The performance sensitivity of the MS in terahertz is excellent. Recently, we have witnessed the phenomenon of us using nanoparticles in many articles to improve efficiency and sensitivity [26,27,28], as can be observed in Figure 9. Because of this phenomenon, we can describe the amount and concentration of toxic particles in an aqueous environment with high accuracy because there is a good and stable correlation between the frequency and amount of toxins in the aqueous environment, as described in Figure 9.

In this work, we present a proof of concept of a new real-time sensor for micropoison detection in aqueous solutions.

This concept led to an innovative idea for creating a terahertz frequency detector, the creation and production of which is efficient and substantially cheaper than any other solution method, such as chemical solutions, that are available today. The physical structure is simple and can be applied to many applications in harsh aquatic environments.

## Figures and Tables

**Figure 1 sensors-22-01279-f001:**
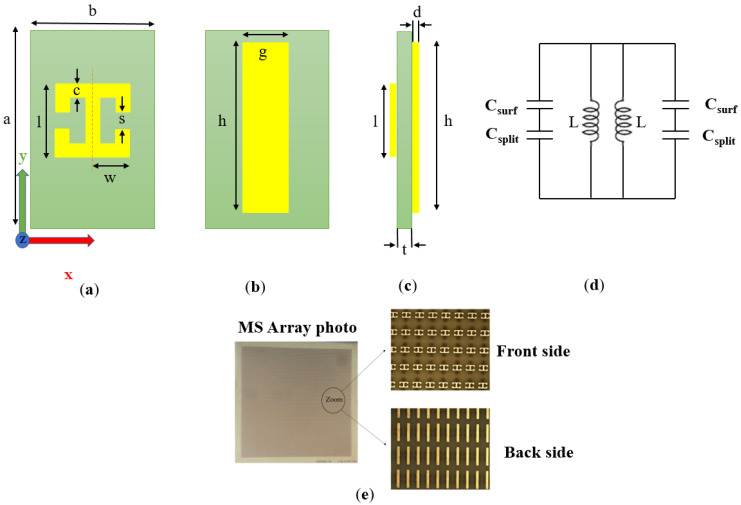
Perfect absorber MS sensor structure. The DSRR at the front side includes axis direction (**a**), cut wire at the back side (**b**), side view of the sensor (**c**), the equivalent circuit of the sensor (**d**), and photo of the MS sensor and microscopic zooming on the front side and back side (**e**).

**Figure 2 sensors-22-01279-f002:**
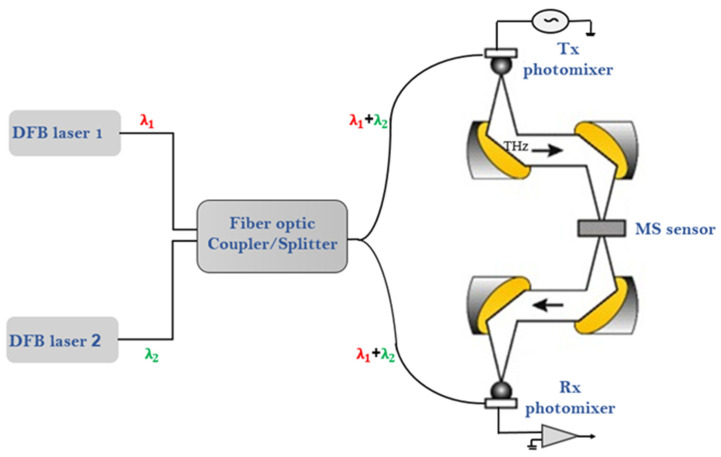
The measurement setup. A Terahertz frequency-domain spectroscopic system with a four-mirror configuration was used to measure the MS sensor transmittance.

**Figure 3 sensors-22-01279-f003:**
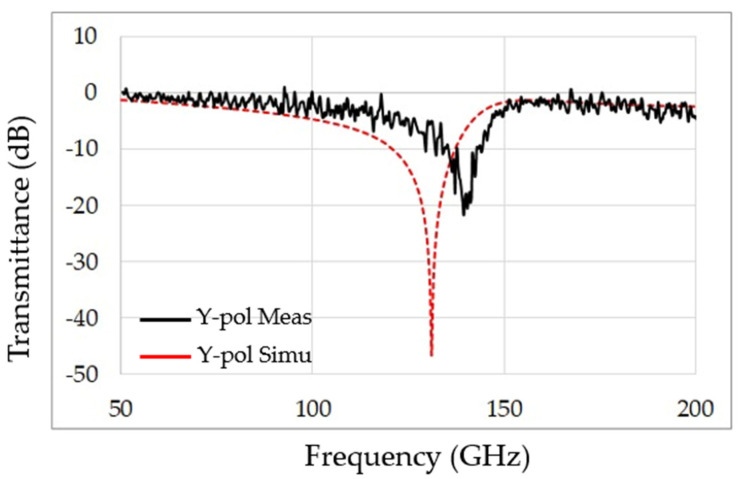
Spectral signature of the perfect absorber measured by a Terahertz frequency-domain spectroscopy system—the simulation and real results.

**Figure 4 sensors-22-01279-f004:**
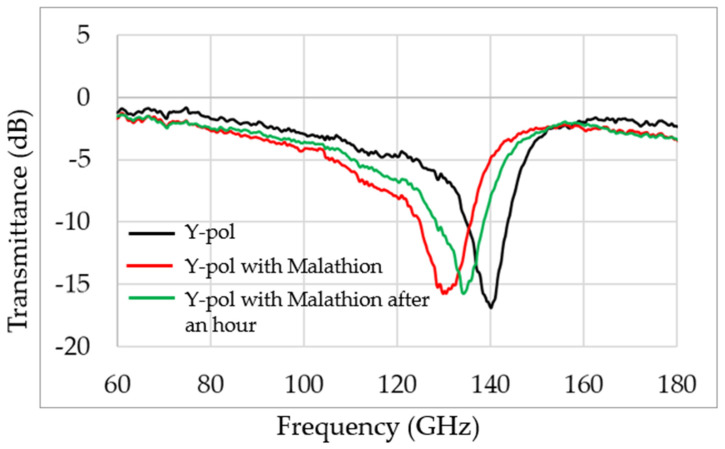
Experimental results of the transmission T. Black line is the pure MS sensor transmission, and the red line and the green line are the transmission measurement immediately after exposure to malathion and one hour after the exposure, respectively.

**Figure 5 sensors-22-01279-f005:**
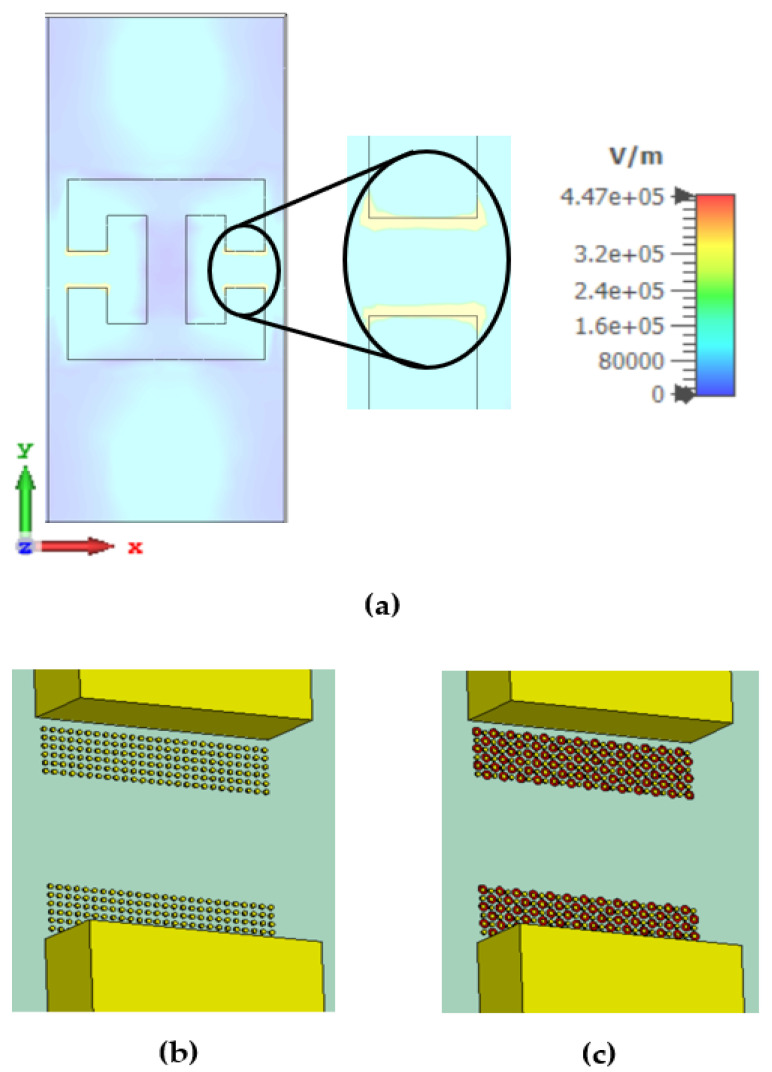
DSRR gap with gold nanoparticles.one unit cell and the electric field intensity distribution on the MS are represent in (**a**), the GNP are represented in the gap of DSRR (**b**), and adhesion of malathion particles to the NGP are represent in figure (**c**).

**Figure 6 sensors-22-01279-f006:**
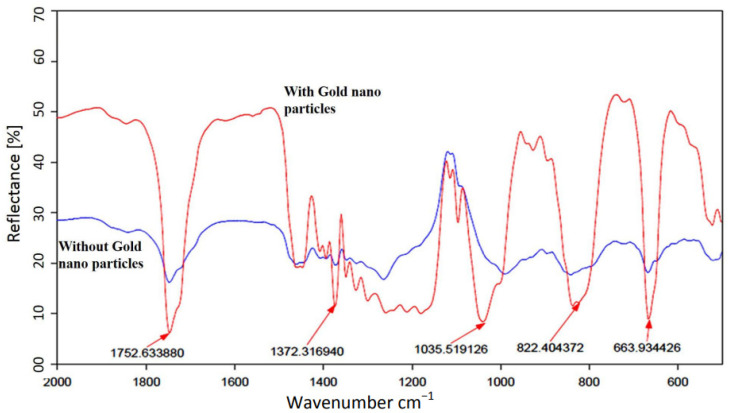
TIR reflection measurements of a dried drop of malathion on silicon substrates for two cases: with a gold nanoparticle coating (red) and without the coating (blue).

**Figure 7 sensors-22-01279-f007:**
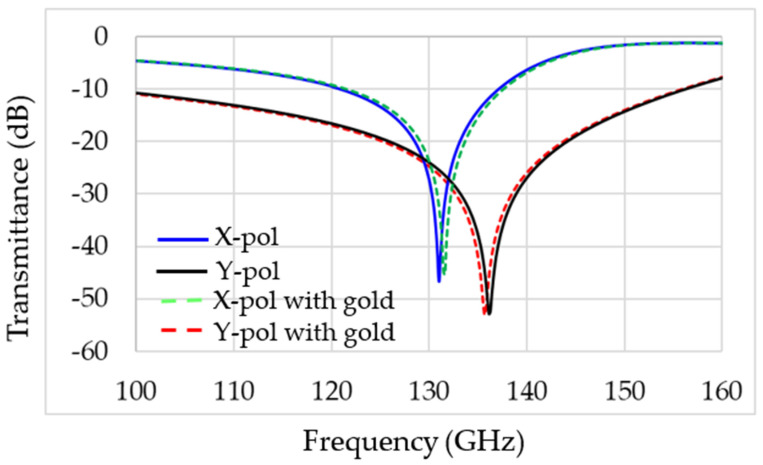
Simulation results for the transmittance of the designed MS. The solid lines are without gold nanoparticles, and the dashed line is with gold nanoparticles in X and Y polarization.

**Figure 8 sensors-22-01279-f008:**
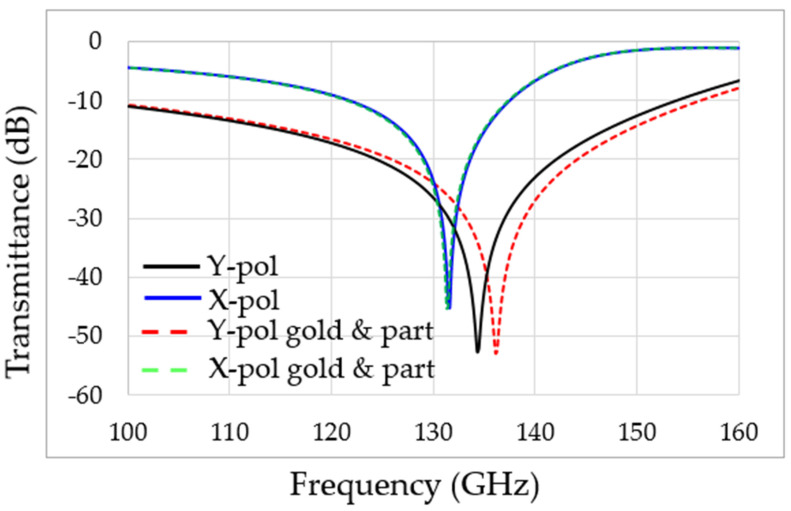
Simulation results for the transmittance of the designed MS. The red dashed line reflects the change in absorption frequency with gold nanoparticles in Y polarization.

**Figure 9 sensors-22-01279-f009:**
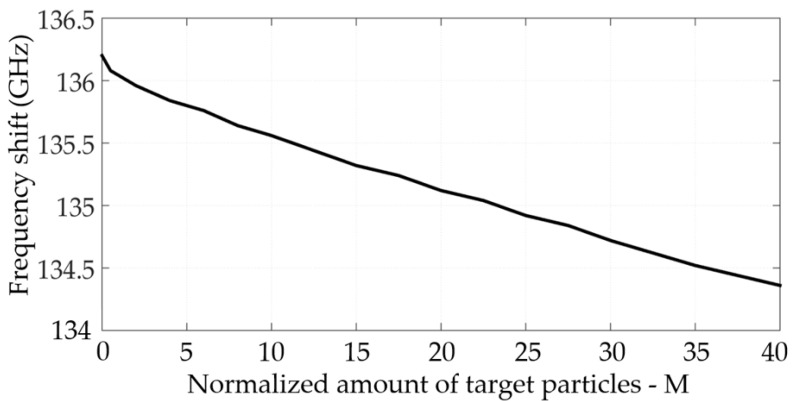
The black line describes the dependence of the absorption frequency on the concentration of toxins in an aqueous environment.

**Table 1 sensors-22-01279-t001:** Unit cell parameters.

Parameter	Description	Value (mm)
a	Unit cell length	1.6
b	Unit cell width	0.6
l	DSRR length	0.5
w	DSRR half width	0.25
c	DSRR strip width	0.1
s	DSRR gap	0.1
g	CW width	0.2
h	CW length	1.2
t	FR4 substrate thickness	0.1
d	Copper thickness	0.035

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
