# Peer review of "Real-Time Metasurface Sensor for Monitoring Micropoisons in Aqueous Solutions Based on Gold Nanoparticles and Terahertz Spectroscopy"

_sensors, 2022, doi:10.3390/s22031279_

Round 1

Reviewer 1 Report

The paper under consideration is a copy/duplication of the author's previous works published in 10.1109/COMCAS.2015.7360466, https://doi.org/10.1117/12.2214085. The design and the explanation of the idea is the exact copy of the previous papers. Even the figures are the same. The paper is written in a poor English language. And the theory to most of the concepts is missing. The figures are of poor quality. The author didn't even try to put any effort to make the proper template of the paper. There is no novelty and will not be useful to the scientific community. Based on my opinion, I am not willing to accept the paper. 

Author Response

First of all, I would like to thank you for reviewing the article.
 Below is our reference

"The paper under consideration is a copy/duplication of the author's previous works published in 10.1109/COMCAS.2015.7360466, https://doi.org/10.1117/12.2214085. The design and the explanation of the idea is the exact copy of the previous papers. Even the figures are the same. The paper is written in a poor English language. And the theory to most of the concepts is missing. The figures are of poor quality. The author didn't even try to put any effort to make the proper template of the paper. There is no novelty and will not be useful to the scientific community. Based on my opinion, I am not willing to accept the paper. "

Says and claims that the article published in 2015 was a promising discovery for a perfect absorber to detection response frequency and identification of toxins.

The current article dramatically improves the detection capability, moreover causes the annexation and adhesive of toxins to metamaterial "Adhesive" surface in the frequency that we Planning then Adhesive is a Upgrade to the perfect absorber because of gold nanoparticles, this phenomenon produces innovation that  not presented in the previous article, the whole matter of innovation is presented in chapter 4 in the article, Discussion line no 141, given that gold nanoparticles are available in industry, and the perfect absorber that represent us simple to fabricate whit good know technology , this means innovation and economic feasibility to create a sensor for the detection and treatment of pollutants in an aquatic environment.

Reviewer 2 Report

Abramovich and coworkers present an interesting sensor design to detect poison in water via the resonance shift of a metasurface. The authors present both numerical and experimental results. Overall, these are nice results and I believe they are suitable in scope for the MDPI Sensors journal.

Besides a few minor remarks, my main concern is the improper use of the term “perfect absorption“ and the missing acknowledgment of key metamaterial literature on that topic. If the authors can satisfactorily improve their manuscript with respect to my concerns, I am happy to recommend publication after another round of review.

Major comments:

The authors neither references seminal works on metamaterial based perfect absorption (PA) (https://doi.org/10.1103/PhysRevLett.100.207402, https://journals.aps.org/prx/abstract/10.1103/PhysRevX.5.031005, https://onlinelibrary.wiley.com/doi/full/10.1002/adfm.202005310) nor do the authors explain what perfect absorption means. I recommend the authors refer to a tutorial paper like https://www.osapublishing.org/aop/abstract.cfm?uri=aop-11-4-892 to clearly see that PA is a scattering anomaly for which a zero of the scattering matrix lies EXACTLY ON (as opposed to merely near) the real frequency axis. In practice, this means that the notch of a perfect absorber should be very deep, for instance, in https://onlinelibrary.wiley.com/doi/full/10.1002/adfm.202005310 the authors achieved below -70dB).

Another issue is related to the tunability of PA. In principle, tuning PA and maintaing its “perfectness“ is a tremendous challenge that was mastered only very recently (https://arxiv.org/abs/2108.06178). Just through tuning a system, the zero drifts away from the real frequency axis.

The authors should properly introduce the concept of PA and its meaning. I also invite the authors to critically question to what extent they observe PA. My understanding is that in simulation they may be seeing PA, but due to fabrication inaccuracies and environmental perturbations, they do not manage to see PA in the experiment (insufficient notch depth). Similarly, upon tuning, PA is not maintained, from what I can see.

Minor comments:

  1. Please provide photograpic images of the experimental setup.
  2. Why does line 32 state “millimeter wavelength“ if the study operates in THz regime?
  3. Why does the measured transmission in Figure 3 exceed unity? Even in the presence of noise, this is very surprising.
  4. Please additionally plot all spectra on a logarithmic scale. The notch depth in dB is crucial to assess how “perfect“ the absorber is.

Author Response

First of all, I would like to thank you for reviewing the article.
Below is our reference.

Major comments:

The authors neither references seminal works on metamaterial based perfect absorption (PA) (https://doi.org/10.1103/PhysRevLett.100.207402, https://journals.aps.org/prx/abstract/10.1103/PhysRevX.5.031005, https://onlinelibrary.wiley.com/doi/full/10.1002/adfm.202005310) nor do the authors explain what perfect absorption means. I recommend the authors refer to a tutorial paper like https://www.osapublishing.org/aop/abstract.cfm?uri=aop-11-4-892 to clearly see that PA is a scattering anomaly for which a zero of the scattering matrix lies EXACTLY ON (as opposed to merely near) the real frequency axis. In practice, this means that the notch of a perfect absorber should be very deep, for instance, in https://onlinelibrary.wiley.com/doi/full/10.1002/adfm.202005310 the authors achieved below -70dB).

We have received the opinion of the reviewer and even facilities and expand information on the perfect absorber.

Given that this article represent domain close to the THz spectroscopy, it is very difficult to produce the perfect absorber in PCB technology because of the small size of the one unit cell.

PCB technology is a manufacturing technology in the electronic components field (printed circuits) that has proven to be efficient, simple and inexpensive to manufacture.

We "stretched" the production limits of the PCB machine, in order to produce absorber for the THZ spectroscopy, beyond the physical limits of the unit cell we will not be able to produce with this technology, which means to product whit more expensive ethnology

We have attached the reference sources of information and articles and even in articles the frequency range is around 3-20 GHz while our absorption frequency is about 140 GHz.

It is also important for us to emphasize that in this article the absorber frequency is dominant in the decision, use perfect absorber is as a detector and absorber nanoparticles create the innovations and not just the perfect absorber, all the superposition of the material properties and the gold nanoparticles are the subject of this article and the simple production and inexpensive capacity.

Another issue is related to the tenability of PA. In principle, tuning PA and maintain its “perfectness“is a tremendous challenge that was mastered only very recently (https://arxiv.org/abs/2108.06178). Just through tuning a system, the zero drifts away from the real frequency axis.

The authors should properly introduce the concept of PA and its meaning. I also invite the authors to critically question to what extent they observe PA. My understanding is that in simulation they may be seeing PA, but due to fabrication inaccuracies and environmental perturbations, they do not manage to see PA in the experiment (insufficient notch depth). Similarly, upon tuning, PA is not maintained, from what I can see.

Minor comments:

  1. Please provide photograpic images of the experimental setup.
  2. Why does line 32 state “millimeter wavelength“ if the study operates in THz regime?
  3. Why does the measured transmission in Figure 3 exceed unity? Even in the presence of noise, this is very surprising.
  4. Please additionally plot all spectra on a logarithmic scale. The notch depth in dB is crucial to assess how “perfect“ the absorber is.
  1. The image has been updated
  2. “millimeter wavelength“ if the study operates in THz regime updated line 32
  3. The noise are represent in fig 3 (the real measurement ) is noise as a result of the TeraScan CW THz spectrometer of Toptica phenomenon machine measuring , enclosing the source of the article and the characterization of the measuring instrument. Line 98-100
  4. All the transmission are updated to dB

Reviewer 3 Report

The manuscript sensors-1544929 submitted as a communication, mainly presents a particular absorber metamaterial proposed for sensing micro poisons in aqueous solutions by the assistance of THz spectroscopy. Please see below a list of comments for the authors:

  1. The authors state “The perfect absorber metasurface was realized by printed metal shapes on both sides of an FR4 substrate, which is very inexpensive” This statement is misleading, THz technology and gold-based nanosensors have been successful to some extent, but they can be costly, time-consuming, and require specialized human resources. Please argue.
  2. The nanoparticles employed were not described neither characterized in the presentation of this report.
  3. The manuscript should clearly describe what the main findings add in respect to comparative publications made in topics related to gold nanoparticles for sensing.
  4. It should be demonstrated that the sample behaves as a metamaterial for the selected wavelength.
  5. Do the behavior of the nanoparticles should be plasmonic or protoplamonic?
  6. Advantages and disadvantages in the use of gold nanoparticles for the development of this real time sensor must be discussed taking into other techniques. You are invited to see for instance the high sensibility exhibited by chaotic sensors doi:10.3390/s19214728 or THz sensors https://doi.org/10.1039/D0RA03116J
  7. It is not clear how can be estimated the resolution and the calibration curve of the system proposed.
  8. The selection of the size and shape of the gold nanoparticles for optimizing the sensing performance must be analyzed.
  9. Please justify the design and size of the absorber metasurface structure proposed in this work.
  10. The main results must present better details with an insight that advances the sensing field for micro poisons in aqueous solutions. Then the results must be confronted with updated publications in order to see the importance of this work.

Author Response

First of all, I would like to thank you for reviewing the article.
 Below is our reference,

The manuscript sensors-1544929 submitted as a communication, mainly presents a particular absorber metamaterial proposed for sensing micro poisons in aqueous solutions by the assistance of THz spectroscopy. Please see below a list of comments for the authors:

  1. The authors state “The perfect absorber metasurface was realized by printed metal shapes on both sides of an FR4 substrate, which is very inexpensive” This statement is misleading, THz technology and gold-based nanosensors have been successful to some extent, but they can be costly, time-consuming, and require specialized human resources. Please argue.

In this article a perfectly executed experiment has been performed which is manufactured by a very inexpensive technology called PCB printing, this technology is well known technology and supports the construction of printed circuits.

Our array cost only a few dollars to fabricate.

The technology is very reliable as well as the gold nanoparticle is currently in the industry and is not expensive.

The advantage of the detector is real time (since the substrate is in the water or liquid environment) and in case of contamination we see the creation of a concentration around the detector and the creation of an adhesive of the toxins to the gold nanoparticle, all the phenomenon is in real time and costing a few dollars to create.

Equally important is the issue of the environment as there is no use of chemicals or pollutants.

  1. The nanoparticles employed were not described neither characterized in the presentation of this report.

The main subject of gold nanoparticles has been extensively brought up in article “Metal Nano Layer Coating for Improving the Detection and Recognition of Micro-Poisons Using Reflection Spectroscopic Measurement" and even quoted line 200-205, in this work the advantages of working into parts and connecting the perfect absorber to gold nanoparticle to create a new concept off sensor,

In any case I will get the enlightenment and add theory the subject.

  1. The manuscript should clearly describe what the main findings add in respect to comparative publications made in topics related to gold nanoparticles for sensing.

The electronics and sensing industry is getting smaller and more accurate, nanoparticle technology is contributing a lot to this industry.

Many examples can be seen in the electronics industry for example " Plastic-Compatible Low Resistance "[22] Low resistance conductors are required to enable the fabrication of high- inductors, capacitors, tuned circuits, and interconnects.

Also in the biological and imaging industry there is a great contribution to quality improvement, evidence has many articles that clearly describe the contribution of nanoparticles to the spectral signature of the material[23][24], articles prove this very clearly.

From this we conclude that the contribution of nanoparticles to the accuracy and quality of the detector will greatly contribute to the creation of value for the sensor for pollutants in a liquid environment.

Al this technology is available and development of ultra-low-cost electronic systems which connects our article to dimensional accuracy and a cheap production

Line 44-48

  1. It should be demonstrated that the sample behaves as a metamaterial for the selected wavelength.

Figure 3 represent certainty the behavior of the Metamaterial piece that we designed and produced and it’s a proven spectral signature of the MS.

The illustration describes the simulation using a CST simulator and the result of the experiment (shown in black line fig 3).

From here it can be seen that it is MS that manufactured by a very inexpensive technology called PCB printing, this technology is well known technology and supports the construction of printed circuits Given the costly results are impressive.

  1. Do the behavior of the nanoparticles should be plasmonic or protoplamonic?

plasmonic

  1. Advantages and disadvantages in the use of gold nanoparticles for the development of this real time sensor must be discussed taking into other techniques. You are invited to see for instance the high sensibility exhibited by chaotic sensors doi:10.3390/s19214728 or THz sensors https://doi.org/10.1039/D0RA03116J

We added a reference regarding nanoparticles to the perfect stamps as well as the article in question was quoted for this article

Line 201 -203

  1. It is not clear how can be estimated the resolution and the calibration curve of the system proposed.

the  estimated the resolution and the calibration curve of the system proposed is depend of all the sources and sensors from Topica system in this case the resolution is 10MHz and all the parameters are represent in reference [20] .

we can see that the resolution is more than 40 that what we needed and its support the sensitivity and frequency change depends of toxic amount, the graph  are represent in fig 9.

  1. The selection of the size and shape of the gold nanoparticles for optimizing the sensing performance must be analyzed.

The size and the shape of gold nanoprticals thickness of the deposited layers were 15 nm and 5 nm of both metals (Gold and Aluminum) it’s the same as we represent in  Amir Abramovich, Alexander Shulzinger, Meir Ochana, David Rotshild, “Metal Nano Layer Coating for Improving the Detection and Recognition of Micro-Poisons Using Reflection Spectroscopic Measurement,” Optics and Photonics Journal, 5, 193-199 (2015).

  1. Please justify the design and size of the absorber metasurface structure proposed in this work.

We created a cell array (one unit cell are showed in figure 1) that creates the perfect absorber material (MM). The substrate material used is well-known FR4, whose dielectric permittivity is er = 4 at 100 GHz, and the thickness is 0.1 mm. the diameter of each DSSR is 0.5 mm × 0.5 mm, and the cut wire size is 1.2 mm × 0.2 mm. the distance between two element is 0.2 mm in x axis and 0.9 in y axis.

The physical dimensions of the array Metamaterial Perfect Absorber are 135 mm × 135 mm, and the total number of cell is 21,690. 

Line no 84-85 and a microscopic photo of the MS is shown in Fig. 1(e). 

  1. The main results must present better details with an insight that advances the sensing field for micro poisons in aqueous solutions. Then the results must be confronted with updated publications in order to see the importance of this work.

We received the comments and even added a number of articles as a reference to the results all described in the previous sections

Round 2

Reviewer 1 Report

I am willing to accept the paper. 

Author Response

First of all, thank you very much for your judgment,

Also, your constructive comments have significantly contributed to the article's improvement.

We re-edited and also edited the references of the article.

Take this opportunity to say thank you again.

Reviewer 2 Report

The authors made an effort to take the previous reviews into account.

Two problems remain:

  1. The reference list is very poorly formatted. The authors should check the details for each single reference. For example, for Ref.[13] the authors mixed together details from two distinct references that I mentioned in my previous report. Each of them should be cited separately and correctly.
  2.  The English of the manuscript needs to be edited extensively. This is not a job for the referee. Please find an editing service. Do not publish your manuscript in its current state of English language.

Author Response

First of all, thank you very much for your judgment,

Also, your constructive comments have significantly contributed to the article's improvement.

We re-edited and also edited the references of the article and we received your comments about the article references.

The reference was re-cited.

Reference [8],[9],[12],[15],[16] marked in red for emphasis.

Take this opportunity to say thank you again.

Reviewer 3 Report

The authors have importantly improved the presentation of their work by confronting their main results with an updated state of the art. All the points raised in the review stage have been clarified, and then, in my opinion, this work can be considered for publication present form.

Author Response

(The authors gave the same response as above.)
